# The Multiple Functions of Fibrillin-1 Microfibrils in Organismal Physiology

**DOI:** 10.3390/ijms23031892

**Published:** 2022-02-08

**Authors:** Keiichi Asano, Anna Cantalupo, Lauriane Sedes, Francesco Ramirez

**Affiliations:** Department of Pharmacological Sciences at the Icahn School of Medicine at Mount Sinai, New York, NY 10029, USA; keiichi.asano@mssm.edu (K.A.); anna.cantalupo@mssm.edu (A.C.); Lauriane.Sedes@mssm.edu (L.S.)

**Keywords:** bone lengthening, dilated cardiomyopathy, fibrillin-1, lens dislocation, Marfan syndrome, osteopenia, thoracic aortic aneurysm

## Abstract

Fibrillin-1 is the major structural component of the 10 nm-diameter microfibrils that confer key physical and mechanical properties to virtually every tissue, alone and together with elastin in the elastic fibers. Mutations in fibrillin-1 cause pleiotropic manifestations in Marfan syndrome (MFS), including dissecting thoracic aortic aneurysms, myocardial dysfunction, progressive bone loss, disproportionate skeletal growth, and the dislocation of the crystalline lens. The characterization of these MFS manifestations in mice, that replicate the human phenotype, have revealed that the underlying mechanisms are distinct and organ-specific. This brief review summarizes relevant findings supporting this conclusion.

## 1. Introduction

Reciprocal interactions between cells and their surrounding extracellular matrix (ECM) are central to the development, homeostasis, and function of all vertebrate organs. Stage- and tissue-specific genetic programs regulate ECM formation and remodeling in response to physiological stimuli or tissue injury. In turn, the assembly of specific matrices influences cell behaviors, establishes tissue boundaries, and specifies physical properties of individual organ systems. The diverse roles of the ECM have been broadly assigned to molecules that self-assemble into higher order tissue structures (architectural proteins) or that modulate cellular activities through the presentation of structural information or soluble biochemical signals (matricellular proteins). There are also ECM molecules that fulfill both structural and instructive roles, for example, the large cysteine-rich glycoprotein fibrillin-1, which is an obligatory component of elastic fibers and is the defective gene product in Marfan syndrome (MFS) [1].

Fibrillin-1 monomers self-assemble into 10 nm-diameter microfibrils through a largely undefined multi-step process that includes the incorporation of additional ECM proteins and interactions with cells via integrin receptors, as well as an association with the elastin polymers, to form the elastic fibers (Figure 1A) [2]. Fibrillin-1 microfibrils are also involved in the extracellular control of transforming growth factorβ (TGFβ) bioavailability by modulating ligand distribution, storage, release, and its presentation to resident cells [3]. TGFβ is secreted as a tripartite latent complex made of the biologically active dimer associated with the pro-peptides, which are, in turn, bound to a latent TGFβ-binding protein (LTBPs 1, 3, or 4). Once secreted, the latent TGFβ complex is tethered to the ECM, in part, via a LTBP interaction with fibrillin-1 microfibrils (Figure 1A). The sequestration into the matrix provides a spatially organized reservoir of latent TGFβ molecules that can be rapidly mobilized through the action of integrins and/or proteases [3].

Irrespective of their individual identity, all fibrillin-1 mutations ultimately lead to a significant decrease of microfibrils in the connective tissue of MFS patients. [4]. Cardinal manifestations of the disease involve the cardiovascular, skeletal, and ocular systems [1]. While thoracic aortic disease (thoracic aortic aneurysms and acute aortic dissection; TAAD) has been long considered to be the major life-threatening manifestation, ventricular dysfunction (dilated cardiomyopathy; DCM) is increasingly recognized as another contributor to increased morbidity/mortality. Disproportionate longitudinal bone overgrowth affecting the limbs, ribs, and spine is the most evident skeletal abnormality, whereas bone loss (osteopenia; OP) remains a controversial finding, particularly in pediatric patients. MFS patients display lens dislocation (ectopia lentis; EL), which can severely impair normal vision and can also cause early cataracts, retinal detachment, and glaucoma. The molecular mechanisms underlying these pleiotropic manifestations of MFS have been characterized in genetically engineered mice that replicate the human disease or that harbor a tissue-specific inactivation of the fibrillin-1 gene (*Fbn1*). Here, we briefly describe these loss-of-function mouse studies that have implicitly revealed distinct tissue-specific contributions of fibrillin-1 microfibrils to organismal physiology. The interested reader is referred to several excellent reviews that summarize, more extensively, the structure and relative expression levels of fibrilin-1, and the biogenesis and organization of normal and mutant microfibrils [5,6,7,8].

## 2. Thoracic Aortic Disease

The aortic matrix is organized to sustain the mechanical stresses imposed by pulsatile blood flow, with elastic and collagen fibers distributing and bearing stress, respectively [9]. Smooth muscle cells (SMCs) in the media provide contractility to the vessel, whereas endothelial cells (ECs) in the intima control vascular tone [9]. Thoracic aortic disease in MFS is characterized by the progressive widening of the aortic root and the proximal ascending aorta that increases the risk for death from an acute dissection and rupture of the vessel wall—i.e., TAAD [10]. The disease is associated with elastic fiber fragmentation (elastolysis), excessive collagen accumulation (fibrosis), dysfunctional ECs and SMCs, and localized inflammatory infiltrates [10]. Ultrastructural studies of aortas isolated from MFS mice have also revealed the loss of fibrillin-1 microfibrils that normally connect SMCs to the elastic fibers in the media, and the ECs to the internal elastic lamina in the intima [9,11]. Current protocols for TAAD management in MFS include the administration of drugs that decrease hemodynamic stress, as well as prophylactic surgery to replace the affected aortic segment [10].

While the characterization of *Fbn1*-null mice has implied the prominent role of fibrillin-1 microfibrils in supporting aortic tissue homeostasis during postnatal life [12], the underlying mechanism is yet to be defined due to controversial findings regarding how fibrillin-1 deficiencies may trigger TAAD formation. Early studies of aortic diseases in mice with non-lethal MFS (*Fbn1^C1039G/+^* mice) concluded that increased TGFβ signaling is a primary consequence of a fibrillin-1 deficiency, triggering the promiscuous activation of matrix-unbound latent complexes [13]. This conclusion was based on the finding that the systemic inhibition of either TGFβ or angiotensin II signaling (via the antagonism of the type I receptor, AT1r) prevented aneurysm formation and reduced the excessive accumulation of phosphorylated (p-) Smad2 in the aorta. More recent investigations using mice with lethal MFS (*Fbn1^mgR/mgR^* mice) showed that a fibrillin-1 deficiency in the aorta actually decreased TGFβ signaling, conceivably by precluding interactions between latent complexes and activators [14]. The finding that the early postnatal deletion of the TGFβ type II receptor in the media of wild-type mice, promoting aneurysm formation, supported the notion that baseline TGFβ signaling is absolutely required to sustain the postnatal increase of systolic pressure and cardiac output [15,16]. In this view, increased p-Smad2 levels in the aneurysmal vessels of MFS mice represents a secondary consequence of fibrillin-1 deficiency, driving unproductive ECM remodeling.

Although medial layer degeneration is the primary driver of TAAD progression, emerging evidence suggests that endothelial dysfunction is an early trigger of arterial disease in MFS. ECs regulate multiple physiological functions, including SMC relaxation/contraction and inflammatory responses, largely through nitric oxide (NO) production [17]. Clinical studies of MFS patients associated a fibrillin-1 deficiency with endothelial dysfunction in the brachial artery [18]. The characterization of *Fbn1^C1039G/+^* mice documented the downregulation of endothelial nitric oxide synthase (eNOS) and impaired SMC contractility, which were mechanistically linked to an increase in oxidative stress [19,20]. Additional work implicated eNOS uncoupling, and reduced NO levels caused by increased reactive oxygen species (ROS) production through a novel TGFβ/NADPH oxidase-4 (NOX4) axis [21,22]. Evidence was also presented regarding the abnormally high levels of inducible NOS (iNOS) in the aortic media of both MFS patients and *Fbn1^C1039G/+^* mice [23,24]. Overall, these findings suggest that impaired eNOS-derived NO production causes ROS activation that, in turn, stimulates iNOS derived NO production in the media. 

## 3. Dilated Cardiomyopathy (DCM)

Myocardial disease and ventricular arrhythmias are additional morbidity/mortality factors in MFS. While cardiac valve disease and the stiffening of the aortic wall can promote DCM formation by imposing a volume overload on the left ventricle, several clinical studies of MFS patients have reported myocardial ventricular dysfunction in the absence of severe valvular pathology [25,26,27]. Intrinsic cardiomyopathy was also identified in both mildly and severely affected MFS mice [28,29]. Importantly, the inactivation of the *Fbn1* gene in cardiomyocytes was shown to be necessary and sufficient to promote DCM formation [28]. By using a combination of genetic and pharmacological interventions, it was further shown that the underlying mechanism involves aberrant myocyte signaling by the mechanosensors AT1r and β1-integrin [28]. These results demonstrate that fibrillin-1 microfibrils, deposited in the myocardial matrix, are an integral component of the force-transmitting network of extracellular, cell surface, and intracellular molecules that modulate cardiac function and homeostasis [30]. 

## 4. Osteopenia (OP)

Bone remodels throughout adult life through a locally coupled process of bone resorption by osteoclasts and bone formation by osteoblasts. Soluble signals released from the ECM during resorption participate in regulating different phases of bone remodeling. TGFβ is the most abundant cytokine sequestered in the bone matrix and is a critical regulator of physiological bone remodeling and repair. The absence of reliable standardized protocols to compare bone mineral density (BMD) between MFS patients and unaffected individuals, and of robust normative data for MFS children, are major contributors to ambiguity. In spite of these limitations, several clinical studies have reported reduced axial and peripheral BMD in adults with MFS, and an increased fracture rate in both adult and pediatric MFS patients [31,32,33]. 

Fibrillin-1 microfibrils accumulate widely in skeletal tissues, including trabecular bone and marrow stroma [34]. An early study correlated OP in young MFS mice with a greater abundance of osteoclasts that also displayed increased bone resorptive activity, largely due to the TGFβ-dependent upregulation of receptor activator of nuclear factor kB ligand (RANKL) production in mutant osteoblasts [35]. This observation was later extended with the demonstration that the N-terminus fragment of fibrillin-1 can also inhibit osteoclastic resorptive activity in vitro by binding RANKL [36]. Mice deficient for fibrillin-1 in all cells derived from limb bud progenitors (*Fbn1^Prx-/-^* mice) were employed to characterize the natural history of OP in the absence of life-threatening cardiovascular complications [34]. These longitudinal analyses correlated progressive bone loss with the premature depletion of mesenchymal stem cells (MSCs) and osteoprogenitor cells, which marrow cell culture experiments associated with improper TGFβ activation [34]. These findings implicated fibrilin-1 microfibrils in delimiting the structural boundaries of MSC niches and in modulating their commitment and differentiation through the regulation of soluble signals. 

## 5. Disproportionate Bone Lengthening

A large clinical study that monitored growth spurts and pubertal skeletal maturation in pediatric MFS patients concluded that their greater average height (relative to the general population) is probably accounted for by poorly controlled, rather than overstimulated, growth [37]. Postnatal bone lengthening progresses under the coordinated control of systemic and local signals that instruct the differentiation of epiphyseal growth plate (GP) chondrocytes. There is also evidence from experiments with cultured bone rudiments that the perichondrium restricts longitudinal bone growth, and that exogenous TGFβ can exert the same inhibitory effect, but only in the presence of an intact perichondrium [38,39]. 

The relative abundance of fibrillin-1 microfibrils in the perichondrium/periosteum, together with the stimulation of bone growth by periosteal resection, were originally used to argue that excessive bone lengthening in MFS may be causally related to the loss of epiphyseal constraint by a structurally impaired perichondrial matrix [40]. A different growth-regulating theory postulated that fibrillin-1 deficiency causes TGFβ hyperactivity with the result of stimulating GP chondrogenesis and, thus, bone lengthening [41]. Correlative evidence supporting this hypothetical mechanism includes the fibrillin-1 modulation of TGFβ bioavailability, microfibril accumulation around GP chondrocytes, and the TGFβ-dependent promotion of chondrogenesis in MFS patient-derived stem cells [1,42,43]. By using a combination of in vivo and ex vivo experiments, we recently demonstrated that a fibrillin-1 deficiency in the perichondrium is causally related to a loss of local TGFβ signaling that causes dysregulated GP chondrogenesis and excessive bone lengthening (Sedes L et al. *manuscript submitted*). 

## 6. Ectopia Lentis (EL) 

Together with aortic aneurysms, EL represents one of the main diagnostic criteria of MFS. Fibrillin-1 is the most abundant component of the zonular fibers that anchor the lens to the ciliary body, in addition to transmitting ciliary muscle-generated forces to change focus [44]. While the identity of the cells producing the zonule proteins is yet to be fully determined, recent data from in situ hybridizations suggest that a subgroup of cells residing in the avascular portion (pars plana) of the non-pigmented ciliary epithelium (NCPE) are responsible for producing fibrillin-1 and other major structural components of the lens-holding fibers [45]. Targeted *Fbn1* inactivation in the NCPE documented the structural role of fibrillin-1 microfibrils in the eye, in addition to replicating the natural history of ocular manifestations in MFS [45]. Fibrillin-1 deficiency in the NCPE was shown to result in smaller and mechanically impaired zonular fibers that eventually ruptured, causing EL in mice. Aging mutant mice developed cataracts as result of lost polarity by the unanchored fibers, which, in a few cases degenerated into glaucoma-like abnormalities [45]. 

## 7. Conclusions

The mouse studies described in this review demonstrate that MFS manifestations are associated with organ-specific disease mechanisms that reflect the multiple specialized functions of fibrillin-1 microfibrils in different connective tissues (Figure 1B). Fibrillin-1 plays a strictly structural/mechanical role in the eye by anchoring the lens in place and transmitting forces generated by the ciliary muscle that change the shape of the crystalline structure. A force-transmitting role is also apparent in the heart, where fibrillin-1 microfibrils around cardiomyocytes are involved in transmitting mechanical signals that are converted intracellularly into biochemical signals promoting workload adaptation. Our limited understanding of TAAD pathophysiology has hampered our efforts to delineate fibrillin-1’s role in the aortic wall. While endothelial dysfunction and TGFβ hyperactivity appear to be prominent primary and secondary consequences of a fibrillin-1 deficiency, respectively, we can only speculate that the ECM defect increases both pressure-induced intramural stress and flow-induced shear stress, with the result of perturbing the ability of resident cells to maintain aortic wall homeostasis. The fibrillin-1 control of MSC activities in marrow niches is instructive, rather than structural, because it is mediated through the regulation of TGFβ bioavailability. In contrast to TGFβ hyperactivity in OP, excessive bone lengthening is associated with loss of TGFβ signaling. Irrespective of organ-specific disease mechanisms, MFS manifestations are also examples of pathological (accelerated) tissue aging [46]. As such, findings gathered from the study of MFS mice are highly relevant to our understanding and management of common diseases of the elderly population. 

## Figures and Tables

**Figure 1 ijms-23-01892-f001:**
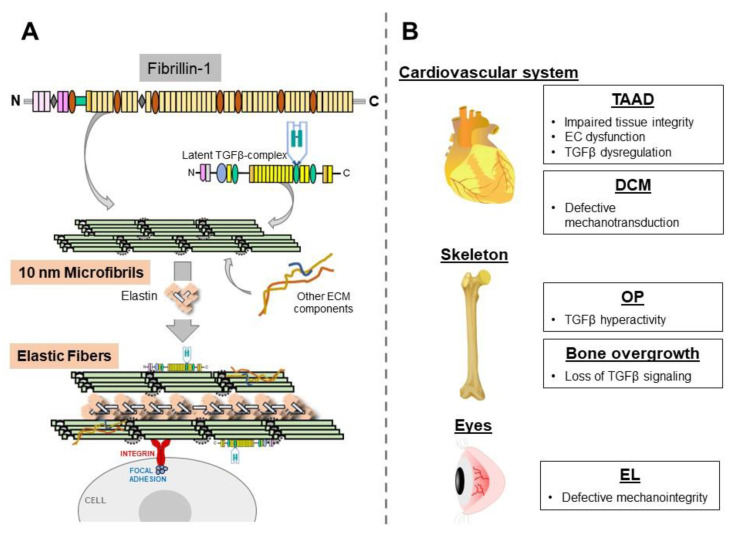
*Panel* (**A**), overall scheme of the main steps in the biosynthesis of 10 nm microfibrils and elastic fibers with indicated interactions discussed in the text. *Panel* (**B**), a summary of the mechanisms underlying selected clinical manifestations characterized in MFS mice.

## Data Availability

Not applicable.

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
