# Peer review of "The Multiple Functions of Fibrillin-1 Microfibrils in Organismal Physiology"

_ijms, 2022, doi:10.3390/ijms23031892_

Round 1

Reviewer 1 Report

Only two minor suggestion to the authors:

please explain all acronyms and

almost an half references are holder than 20 years. I should suggest to consider the possibility of reducing or updating the bibliography.

Author Response

  1. All acronyms have been explained in the abstract, main text and figure legend.
  2. References before the year 2000 were reduced to 5 out of 46 by deleting the historical perspective at the beginning of the Conclusion.

Reviewer 2 Report

This short review article describes the known associations of fibrillin-1 protein with various clinical manifestations related to Marfan syndrome. Overall, this paper is a clinical review of Marfan syndrome with references to preclinical models using knockouts of fibrillin-1. This is an interesting topic, of particular interest to the study of the role of specific ECM proteins in human disease. However, re-structuring of the review and inclusion of relevant figures and references are needed throughout. Specific comments are described below.

-Given the title, the reader expects a review of fibrillin-1 protein, though no descriptions or figures are included to describe protein structure, relative expression patterns in different tissues, and common mutations associated with it in the context of Marfan syndrome.

-Please include the relative abundance of fibrillin-1 in the matrix relative to other ECM proteins, such as collagen type 1, elastin, and surrounding glycosaminoglycans (decorin, lumican, etc).

-Lack of figures and appropriate references throughout heavily limits the quality of the manuscript. (Ex: lines 47-58 - statistics for the relative abundance are included (60-80%) without any reference to a primary article; mention of animal models again without any accompanying references).

-Inclusion of a figure summarizing the anatomical descriptions and their associated pathology related to fibrillin-1 loss would be beneficial.

-Better descriptions including appropriate references to the literature regarding the mechanisms underlying how loss of fibrillin-1 affects microfibril structure is needed.

-Overuse of arbitrary acronyms throughout is not helpful to the reader, especially not as section headers. Please limit acronyms to essential or standard usages given the general audience of the journal. Also, please define all acronyms at first mention (RGD, TGFbeta, etc).

-Please correct all grammatical errors throughout (ex - line 41, laten).

Author Response

  1. As we stated, the focus of our review was to discuss articles that, by using MFS mice, have provided insight into molecular mechanisms underlying selected disease manifestations. To avoid further confusion, we have restructured the Introduction and referenced more exhaustive summaries related to the fibrillin-1 and MFS topics suggested by the Reviewer (see closing two statements of the Introduction).
  2. Please see the answer above.
  3. A figure depicting the principal steps of microfibril/elastic fiber biogenesis and summarizing the mechanistic findings discussed in the review has been added.
  4. Please see the answer above.
  5. All acronyms have been explained in the abstract, main text and figure legend.
  6. Typos have been corrected.

Reviewer 3 Report

Mutations in fibrillin-1(Marfan syndrome) is responsible of thoracic aortic aneurysm, myocardial dysfunction, progressive bone loss, disproportionate skeletal growth and dislocation of the crystalline lens. The present brief review describes how studies carried out in mice have contributed to decipher the underlying mechanisms of those organ-specific disfunctionings. Molecular mechanisms underlying these pathologies have been characterized in genetically engineered mice to replicate human diseases characterized by a tissue-specific inactivation of the fibrillin-1gene. Therapeutic implications and future potential researches are discussed convincingly. This review is a significant contribution for the interdisciplinary scientific community involved in this specific field. It is very well written and the various topics are very clearly and consistently presented. It is easy for the reader going through the paper and following the discussion.

Author Response

We truly appreciate the laudatory comments of the reviewer

Round 2

Reviewer 2 Report

The revision includes an added figure showing fibrillin protein interactions with latent TGFb and a panel showing very general pathological effects associated with MFS. Added mention of the relative abundance of fibrillin in the ECM relative to collagen is really needed since the entire review focuses on the clinical manifestations associated with defects in ECM structure per fibrillin loss, as mentioned in the previous review. It is not clear what other changes were made in the revision. Minor topographical issues are still present (line 44; lines 46, 69 (repeated TAAD explanation), etc).
